# Effects of Pelvic-Tilt Imbalance on Disability, Muscle Performance, and Range of Motion in Office Workers with Non-Specific Low-Back Pain

**DOI:** 10.3390/healthcare11060893

**Published:** 2023-03-20

**Authors:** Won-Deuk Kim, Doochul Shin

**Affiliations:** 1Department of Physical Therapy, Graduate School of Kyungnam University, Changwon-si 51767, Republic of Korea; 2Department of Physical Therapy, College of Health Science, Kyungnam University, Changwon-si 51767, Republic of Korea

**Keywords:** low-back pain, risk factors, range of motion, hip joint, knee joint

## Abstract

Imbalance in the pelvic tilt is considered to be a major variable in low back pain. The purpose of this study was to investigate the effects of pelvic-tilt imbalance on trunk- and hip-muscle performance, range of motion, low-back pain, and the degree of disability in office workers. This was a cross-sectional study conducted in a physical therapy clinic on forty-one office workers diagnosed with non-specific low-back pain. Among the office workers with non-specific low-back pain, 25 were assigned to the pelvic-tilt-imbalance group and 16 to the normal group without pelvic-tilt imbalance. In order to determine the differences according to the imbalance in pelvic tilt, the pain intensity and disability indices were compared between the groups. In addition, the muscle performance and range of motion of the trunk and hip joints and foot pressure were measured and compared. There were differences between the groups in the disability indices and the ratio of internal and external rotation of the hip joint. However, there were no differences in the other variables. Pelvic-tilt imbalance in office workers with non-specific chronic low-back pain may exacerbate the degree of disability and be related to hip-joint rotational range of motion.

## 1. Introduction

Non-specific low-back pain refers to pain around the lower back and buttocks that appears without pathological anatomical causes found in various medical tests [1]. More than 90% of the population experience low back pain at least once in their lifetime, and 90–95% of patients are diagnosed with non-specific low-back pain [2]. Low-back pain recurs easily [3]. Bergquist et al. [4] reported that after 1 year of follow-up of patients with acute low-back pain, 65% experienced additional pain at least once. Considering the high incidence of low-back pain, many people experience it, and do not recover easily due to the high recurrence rate.

It is reported that various factors, such as age, sex, muscle strength, flexibility, physique, and weight affect the risk of non-specific low-back pain. Among various risk factors, it was reported that trunk-muscle function, hip-joint-muscle strength, the range of motion of the trunk and lower extremities, and the alignment of the pelvis greatly contribute to low back pain [5,6,7]. In addition, weakness in the abdominal, paraspinal, and multifidus muscles is found in most patients with non-specific low back pain [8,9].

In a systematic review conducted by Hamberg et al., a significant correlation was reported between trunk-muscle strength, spinal mobility, and risk of low-back pain [10]. If the muscle strength of the hip joint is weakened, this can cause problems with balance and posture control during activities, which can act as a risk factor for low-back pain [11,12]. In addition, when the hip joint lacks flexibility, compensatory movements occur in the lumbopelvic region, and microtrauma to the lumbar vertebrae causes low-back pain [13,14]. Kim et al. [15] reported that lumbar flexion and posterior pelvic tilt significantly increased in subjects with low back pain when hip flexion was performed in the sitting position. Sadeghisani et al. [16] reported that the rotation of the lumbar spine and pelvis was greater when the subjects with low back pain performed hip rotation. 

The pelvis is a structure that connects the lower extremities and trunk and plays a role in controlling the balance of force between the upper and lower body [17,18]. This change in pelvic tilt affects the length of the muscles around the spine and hip joint [19,20]. For this reason, a change in pelvic tilt creates abnormal alignment of the spine, which causes great stress on the lumbar vertebrae [21]. Krol et al. [22] reported that subjects with low-back pain had a greater anterior pelvic tilt than subjects without back pain, and that there was a significant relationship between anterior pelvic tilt and lumbar lordosis. Cejudo et al. [23] reported that there was a significant relationship between hamstring flexibility and recurrent back pain, pelvic tilt, and lumbar curve. As a result of a comprehensive analysis of 24 studies in a systematic review by Yoon WG et al., it was found that there was a significant relationship between left and right pelvic imbalance and back pain [24]. Several clinical studies have described the relationship between changes in the lumbar curve and those in the trunk and hip muscles according to the anteroposterior tilt of the pelvis. However, studies on the relationship between various physical risk factors according to the left–right imbalance of the pelvis are lacking. Therefore, in this study, the relationship between the presence of pelvic-tilt imbalance and various physical risk factors for low-back pain was investigated.

## 2. Materials and Methods

### 2.1. Participants

This study was conducted on office workers at the S Company Musculoskeletal Therapeutic Exercise Center in Seoul, Korea. The study analyzed data collected from May 2018 to December 2018 for the health of workers at the Center. Subjects were recruited through the internal bulletin board. The participants, who were office workers diagnosed with non-specific low back pain by doctors, were enrolled into two groups; 25 with a difference in pelvic tilt and 16 without a difference in pelvic tilt. Among the office workers who participated in this study, those who performed simple data-entry tasks, those who stood for long periods of time, those who used a computer for less than 20 h per week, those who were taking drugs for back pain, those who had a numeric pain rating scale (NPRS) score of 2 or less, and those with Oswestry Disability Index (ODI) scores of less than 10 points were excluded. The procedure and purpose of the study were fully explained to all the participants, who signed an informed consent form. All research was conducted with the approval of the Kyungnam University Research Ethics Committee (1040460-E-2021-003).

### 2.2. Research Procedures

This study was conducted to investigate the differences between non-specific low-back-pain symptoms and physical characteristics according to the presence or absence of pelvic tilt in participants with non-specific low-back pain. The symptoms of patients with non-specific low-back pain were investigated based on the pain intensity and disability index. The NPRS was used to measure pain intensity, and the ODI was used as a disability index. 

To measure the pelvic alignment of the subjects, a surface-topography device, Formetric 4D (Diers Medical Systems, Chicago, IL, USA) was used. The physical characteristics of office workers with non-specific low-back pain were assessed by measuring trunk-flexor and -extensor strength, hip-extensor strength, and trunk-flexor and -extensor endurance. Further, the joint range of motion was measured by trunk flexion and extension, hip flexion and extension, and hip-joint internal and lateral rotation. Erector-spinae-muscle activity and foot-pressure distribution were also measured. The variables for the participants’ physical characteristics were measured three times and the average value was used. Additionally, for the daily lives of the study participants, the weekly computer-usage time and weekly exercise time were investigated. All evaluations were conducted by a physical therapist with 8 years of clinical experience, regardless of the study design.

### 2.3. Pelvis Alignment Imbalance

To measure pelvic-alignment imbalances in office workers with non-specific low-back pain, a Formetric 4D device was used [25]. This device depicts a model of the vertebrae by collimating and shining light across the posterior trunk surface and digitally reconstructing the subject’s posterior surface based on the distortion of these lines. To measure their pelvic tilt, each subject was placed in a standing position in front of the Formetric 4D device, with the posterior iliac spine of the back and pelvis visible. Imbalance in pelvic alignment refers to an inclination of the pelvis in the coronal plane, and the left and right pelvic heights of the participants were different. 

### 2.4. Numeric Pain-Rating Scale (NPRS)

Pain intensity was measured using a numerical pain-rating scale. The study participants selected pain intensity from 0 to 10, with 0 indicating no pain and 10 indicating the most severe pain [26].

### 2.5. Oswestry Disability Index (ODI)

Low-back-pain-disability index was assessed using the ODI. The ODI consists of 10 items: pain level, lifting objects, personal hygiene, walking, standing, sitting, sleep, social activities, sex life, exercise, and travel. This study evaluated nine items, excluding questions related to sexual life, which are personally sensitive. On a 6-point scale for each item, the score ranged from 0 to 5 and was converted to percentage form with 0–20% meaning mild disability, 21–40% moderate disability, 41–60% severe disability, and >60% severe disability. The KODI is a reliable evaluation, with Cronbach’s α = 0.85 [27].

### 2.6. Muscle Power of the Lumbar Flexors, Extensors, and Hip Extensors

To measure the strength and ratio of the lumbar flexor and extensor muscles, a device (HUR 5310, HUR, Finland) was used to measure the muscle strength in the isometric state while the subject pushed a bar in the sitting position. To measure the lumbar-flexion strength, the subject approached a bar located in front of their chest in an upright sitting position, held it with both hands, pushed it, and bent the body strongly forward. The lumbar-extension-muscle strength was measured by the subject pushing the bar located on their back in a sitting position while lying down, and the unit of measurement was kg. The muscle strength of the lumbar-spine extensors was measured while each participant lay on their back with the bar positioned behind their back in an upright sitting position, and the unit of measurement was kg. The ratio of the lumbar flexor and extensor muscles was calculated by dividing the lumbar-flexor-muscle strength by the lumbar-extensor muscle strength.

To measure hip-extensor strength, the subjects were asked to flex the knee joint to 90 degrees in the prone position and extend the hip joint without other compensatory motions. To measure hip-extension-muscle strength, a digital muscle-strength-measurement device (JTEC Medical Commander Muscle Tester, JTEC Medical, Midvale, UT, USA) was placed in the center of the posterior part of the thigh closest to the knee joint.

When the subject performed the movement, the pelvis was fixed so that other compensatory effects (anterior pelvic tilt, hip rotation, hip abduction, etc.) did not occur. The maximum muscle strength was measured three times, and the average value was used. A 30-s break was provided between each measurement [28]. To calculate the left and right asymmetry of the maximum voluntary isometric contraction (MVIC) of the hip-joint extensor, the smaller value was divided by the larger value of the left/right muscle strength.

### 2.7. Muscle Endurance of Lumbar Flexors and Extensors

To measure the endurance of the lumbar-flexor muscle, the subject was instructed to adopt a stance in which the hip was bent by 45 degrees and the knee by 90degrees in a supine position, hold the back of their neck, and lift their upper body with both hands, while keeping both shoulder blades as far apart as possible. The time taken was measured using a stopwatch.

To measure the endurance of the extensors of the lumbar spine, in the prone position, a pillow was placed on the lower abdomen to reduce lumbar lordosis and to maintain the upper body as high as possible until the sternum was off the floor [29].

### 2.8. ROM in Lumbar Flexion and Extension

The joint range of motion of lumbar flexion and extension was measured using a digital dual inclinometer in standing position (JTECH Dual Inclinometer, JTECH Medical, USA). The inclinometer was placed on the T12 spinous process and the S1 spinous process; in the upright position, with the knee straight, trunk flexion and extension were performed, respectively, and measurements were made [30].

### 2.9. ROM in Hip Extension, Flexion, and Internal and External Rotation

A modified Thomas test was used to measure the hip-extension range of motion. The subject was placed on the edge of the bed with the hip joint and knee bent, and the shin of the opposite leg to be tested was held and fixed with both hands. While the hip joint and knee flexion were maintained, each participant was instructed to slowly extend the hip joint, and the hip-joint-extension angle was measured using a digital inclinometer (JTECH Dual Inclinometer, JTECH Medical, USA) in the lowest possible position. 

To measure the hip-joint-extension angle, the inclinometer was placed in front of the femur between the center of the greater trochanter of the femur and the lateral condyle [31]. In order to measure the hip-joint-bending range of motion, the measurer manually flexed the hip joint and knee of the subject as much as possible in the supine position and measured them using a goniometer. To measure the angle of hip flexion, the center of the goniometer was placed on the greater trochanter, the fixed arm was placed parallel to the horizontal plane, and the movable arm was placed to the side [32]. 

Internal rotation and lateral rotation of the hip joint were measured in the end range, where the subject’s hip joint was manually rotated internally and laterally by the evaluator in a position with the knee bent 90 degrees. To measure hip rotation, a digital inclinometer (JTECH Dual Inclinometer, JTECH Medical, USA) was placed at the distal end of the fibula, excluding the lateral condyle [33]. The left–right asymmetry in hip flexion, medial rotation, and lateral rotation were calculated by dividing the small value by the large value for each.

### 2.10. ROM in Knee Flexion and Extension

A digital inclinometer (JTECH Dual Inclinometer, JTECH Medical, USA) was used to measure the range of motion in flexion and extension of the knee joint. To measure the knee-joint-extension range of motion, the subject flexed their hip joint by 90 degrees in the supine position. The researcher placed the inclinometer at the distal end of the tibia and manually lifted the lower leg. The angle of the inclinometer was measured at the angle at which the participant’s knee could no longer be extented or at which the subject felt pain [34]. 

To measure the knee-joint-flexion range of motion, the subject was placed in a prone position. The measurer fixed the pelvis to maintain a neutral position, placed the inclinometer at the distal end of the tibia, and manually lifted the lower leg. The angle of the inclinometer was measured at the angle at which the participant’s knee could no longer be bent or at which the subject felt pain [34].

### 2.11. Trunk-Muscle Activity and Foot Pressure

To measure the muscle activity of lumbar extension, electrodes were attached to the erector-spinae muscles on both sides of L3 and L5 while maintaining knee extension, bending the lumbar vertebrae as comfortably as possible in the standing position, and returning to the upright position. This method was used to measure afferent and efferent activities of the lumbar-spine extensors. To calculate the centripetal and efferent left–right symmetry of maximal activity of the lumbar-spine extensors, small values of left–right muscle activity were divided by large values. Muscle-activity-measurement equipment (ScanVision PLUS [+] sEMG, MyoVision, Lynnwood, WA, USA) was used for measurements [35].

Foot pressure was measured using a Pedoscan (Diers Medical Systems, Chicago, IL, USA). The device captures and displays the pressure distribution applied to the feet of a person while standing. After measuring the subject’s foot pressure using this equipment, the subject’s left–right-foot-pressure ratio and the front- and rear-foot-pressure ratios of the left and right feet were calculated. 

### 2.12. Statistical Analysis

General characteristics, such as disability index, pain intensity, computer-use time, and exercise time, were compared according to the pelvic tilt of participants with non-specific low-back pain. The power and endurance of the lumbar flexors and extensors, and the power and endurance of the hip extensors, were compared. In addition, the ROM of the trunk, hip joint, and knee joint, the muscle activity of the erector spinae, and foot pressure were compared. Statistical analyses were performed using SPSS 21.0. The Kolmogorov–Smirnov test was used to assess data normality. An independent t-test was used to compare the groups according to the presence or absence of pelvic tilt.

## 3. Results

A total of 61 subjects were recruited in this study, and 41 subjects eventually participated, excluding those who met the exclusion criteria and dropouts. The patients were divided into 25 in the experimental group with pelvic tilt and 16 in the control group without pelvic tilt. The characteristics of the experimental group were as follows: age, 32.72 ± 5.72 years; weight, 68.16 ± 12.2 kg; height, 170.86 ± 8.26 cm; body mass index (BMI), 23.08 ± 3.03; and difference in pelvic tilt, 5.8 ± 3.38 mm. For the control group, these values were as follows: age, 32.63 ± 5.85 years; weight, 64.88 ± 7.99 kg; height, 169.33 ± 5.74 cm; BMI, 22.96 ± 1.61; and difference in pelvic inclination, 0 mm (Table 1). There was a significant difference between the groups with regards to the low-back-pain-disability index, with 17.84 ± 6.71 points in the experimental group and 10.13 ± 4.22 points in the control group (Table 2). Regarding the differences in the range of motion of the trunk, hip, and knee joints between the groups, the hip-joint-internal-rotation ratio was 0.68 ± 0.09 in the experimental group and 0.86 ± 0.16 in the control group. There was a significant difference of 0.85 ± 0.16. Moreover, there was a significant difference in knee-flexion ratio: 0.94 ± 0.6 in the experimental group and 0.96 ± 0.4 in the control group (Table 3). The pain intensity, computer-use time per week, exercise time per week (Table 2), trunk-flexor- and -extensor-muscle strength, hip-extensor-muscle strength (Table 4), trunk flexion and extension, hip internal rotation, lateral rotation, hip-flexion–extension range of motion, hip-joint flexion–extension ratio, knee-joint flexion–extension range of motion, knee-joint-extension ratio (Table 3), trunk-muscle activity, and foot-pressure distribution were not significantly different (Table 5).

## 4. Discussion

This study aimed to compare the risk factors for low-back pain according to differences in pelvic imbalance. The subjects participating in this study were divided into an experimental group and a control group according to these differences. The pain intensity, low-back-pain-disability index, weekly computer use, weekly exercise time, trunk and lower-extremity-muscle strength, joint range of motion of the trunk and lower extremities, trunk-muscle activity, and foot-pressure distribution were measured.

The results showed that the low-back-pain-disability index was significantly higher in the experimental group than in the control group. Chuang et al. [36] reported that in a study analyzing the risk factors of low-back pain in patients with or without degenerative discs, there was a significant correlation between pelvic-tilt change and the low-back-pain-disability index in both groups. Salt et al. [37] reported a positive correlation between increased pelvic tilt, pain, and disability index in a study that followed up the pain and disability indices of patients with low-back pain for 6 months. In this study, it was also found that the subjects with different pelvic imbalances had higher low-back pain disability indices. This result is thought to have been due to the abnormal alignment of the spine due to the change in pelvic imbalance, which created muscle imbalances in the trunk and lower extremities. 

For the joint range of motion, between the groups, the ratio of internal and lateral rotation of the hip joint was significantly decreased. Rikin et al. [38] investigated the degree of hip-joint internal rotation according to changes in the pelvic tilt. As a result, when the pelvis was tilted anteriorly, the hip-joint internal rotation decreased, and when the pelvis was tilted posteriorly, the internal hip-joint rotation increased. Kisuke et al. [39] reported that when the anterior-pelvic-tilt angle increased by 1 degree, the anterior coverage of the femoral head increased by 0.5%, and the posterior coverage decreased by 0.3%. In addition, when the anterior pelvis tilts, the iliopsoas-muscle tone increases, and when the posterior pelvis tilts, the hamstring shortens and its muscle tone increases [40,41]. The reason why the difference in left and right pelvic tilt in this study changed the hip-joint-rotation ratio is thought to be because the difference in the tilt of the left and right pelvis changed the femoral head coverage and also changed the length and tone of the muscles around the hip joint. 

As a result of examining the knee-joint-flexion and -extension range of motion between the groups, we found that there was a significant difference in the knee-joint-flexion ratio. Jeon et al. [42] reported a significant correlation between pelvic tilt, knee-extension–flexion range of motion, and hamstring flexibility. In this study, it is thought that the difference in the inclination of the pelvis affects the range of motion in the extension and flexion of the knee, which results in differences in the flexion ratio of the knee joint.

There were no significant differences in the strength and activity of the trunk-flexor and -extensor muscles and hip-extensor-muscle strength between the two groups. These results are thought to have been due to the lack of correlation with the variables measured in the sagittal plane, such as flexion and extension strength and muscle activity, because the subjects in this study were classified based on differences in left and right pelvic tilt in the frontal plane. In addition, contrary to the hypothesis of this study, there were no significant differences in trunk-muscle activity or foot pressure between the two groups. Differences between the left and right pelvic tilt can affect the mobility and balance of the trunk and lower extremities [43]. However, this is not determined only by the tilt of the pelvis and is balanced by various compensatory actions [44]. For this reason, it is thought that there were no differences between the two groups in trunk-muscle activity or plantar-pressure distribution. There were no significant differences between the groups in the weekly computer-use time or the weekly exercise time. It is thought that these results appeared because the weight distribution was not considered when using a computer and the specific types of exercise were not investigated. Future studies will need to improve this aspect.

Regarding the effect size of the statistically significant variables (simple differences only) according to the presence or absence of pelvic imbalance, the hip-internal-rotation ratio was the largest at 1.7, and the knee-flexion ratio was the smallest at 0.39, as shown below.

The hip-external-rotation ratio was found to have an effect size of 0.69 and the ODI was found to have an effect size of 1.38. The results of this study suggest that among the statistically significantly different variables, the hip-internal-rotation ratio is most affected by pelvic imbalance. The ODI also showed that the effect size was large, depending on the presence or absence of pelvic imbalance, suggesting that imbalances in pelvic alignment in office workers with non-specific chronic low-back pain can significantly affect their disability level.

A change in pelvic tilt changes the alignment of the hip joint and spine, which is a risk factor for low-back pain, and changes the range of motion of the hip joint. Therefore, it can be said that the results of this study provide an important index for the evaluation of patients with low-back pain. However, in this study, there was a limitation in comparing the pelvic tilt with various physical risk factors because only the left–right difference in the frontal plane was measured; the anteroposterior inclination difference in the sagittal plane and the pelvic rotation in the horizontal plane were not measured. Future research should be conducted by grouping more diverse methods.

## 5. Conclusions

Office workers often suffer from low-back pain and pelvic tilt because they often sit for long periods of time. Pelvic-tilt imbalance in office workers with non-specific low-back pain can affect their hip-rotation range of motion and degree of disability due to low-back pain. Therefore, the evaluation and treatment of pelvic alignment may be necessary for low-back pain in office workers with non-specific low-back pain. Differences in pelvic malalignment were related to physical risk factors for low-back pain. In particular, the disability index and hip-joint-rotation ratio were fund to affect low-back pain.

## Figures and Tables

**Table 1 healthcare-11-00893-t001:** General characteristics of the subjects.

	With Pelvic-Tilt Imbalance (*n* = 25)			Without Pelvic-Tilt Imbalance (*n* = 16)			*p*
Age (years)	32.72	±	5.72	32.63	±	5.85	0.959
Weight (kg)	68.16	±	12.2	64.88	±	7.99	0.835
Height (cm)	170.86	±	8.26	169.33	±	5.47	0.952
BMI (score)	23.08	±	3.03	22.96	±	1.61	0.878
Pelvic-tilt imbalance (mm)	5.8	±	3.38	0	±	0	0.000

Values are mean ± SD, BMI: body mass index.

**Table 2 healthcare-11-00893-t002:** Comparison of disability, pain intensity, computer-use time, and exercise time between groups.

	With Pelvic-Tilt Imbalance (*n* = 25)			Without Pelvic-Tilt Imbalance (*n* = 16)			*p*
ODI (score)	17.84	±	6.71	10.13	±	4.22	0.000
NPRS (score)	4.2	±	1.58	3.69	±	1.49	0.361
Computer-use time (hour/week)	40.96	±	13.01	38.31	±	20.01	0.843
Exercise time (hour/week)	2.98	±	2.0	3.54	±	3.6	0.947

Values are mean ± SD, ODI: Oswestry disability index, NPRS: numeric pain-rating scale.

**Table 3 healthcare-11-00893-t003:** Comparison of trunk, hip, and knee-joint range of motion between groups.

		With Pelvic-Tilt Imbalance (*n* = 25)			Without Pelvic-Tilt Imbalance (*n* = 16)			*p*
Trunk	Flexion (°)	82	±	31.76	90.06	±	37.05	0.347
	Extension (°)	25.08	±	13.33	27.38	±	14.16	0.781
Hip	Rt. internal rotation (°)	43.32	±	10.77	37.38	±	13.05	0.188
	Lt. internal rotation (°)	33.32	±	11.12	36.13	±	9.24	0.259
	Rt. external rotation (°)	33.08	±	9.0	40.19	±	13.32	0.139
	Lt. external rotation (°)	40.68	±	10.04	40.5	±	9.75	0.663
	Rt. flexion (°)	125.04	±	23.07	123.88	±	19.83	0.926
	Lt. flexion (°)	126.6	±	21.04	125.69	±	18.2	0.905
	Rt. extension (°)	20.8	±	7.68	18.94	±	5.07	0.404
	Lt. extension (°)	20.00	±	7.82	19	±	5.73	0.781
	Internal-rotation ratio	0.68	±	0.09	0.86	±	0.12	0.000
	External-rotation ratio	0.74	±	0.16	0.85	±	0.16	0.016
	Flexion ratio	0.96	±	0.03	0.95	±	0.05	0.905
	Extension ratio	0.79	±	0.17	0.87	±	0.12	0.101
Knee	Rt. flexion (°)	124.44	±	20.85	123.75	±	15.95	0.500
	Lt. flexion (°)	124.8	±	18.27	126.44	±	16.09	0.761
	Rt. extension (°)	149.8	±	18.67	152.63	±	16.89	0.500
	Lt. extension (°)	150.04	±	19.19	155.94	±	18.16	0.517
	Flexion ratio	0.94	±	0.06	0.96	±	0.04	0.032
	Extension ratio	0.97	±	0.03	0.96	±	0.04	0.588

Values are mean ± SD, Rt: right, Lt: left.

**Table 4 healthcare-11-00893-t004:** Comparison of trunk and hip-extensor power and endurance between groups.

	With Pelvic-Tilt Imbalance (*n* = 25)			Without Pelvic-Tilt Imbalance (*n* = 16)			*p*
Lumbar-flexor power (kg)	73.8	±	37.11	66.06	±	38.05	0.534
Lumbar-extensor power (kg)	91.88	±	50.31	83.88	±	43.45	0.761
Lumbar-flexor endurance (sec)	68.96	±	40.47	60.43	±	22.58	0.968
Lumbar-extensor endurance (sec)	89.44	±	58.87	87.31	±	39.74	0.741
Lt.-hip-extensor power (kg)	35.76	±	14.85	35.12	±	14.42	0.864
Rt.-hip-extensor power (kg)	38.46	±	16.99	39.16	±	19.72	0.843
Hip-extensor-power ratio	0.84	±	0.13	0.86	±	0.11	0.802

Values are mean ± SD.

**Table 5 healthcare-11-00893-t005:** Comparison of trunk-muscle activity and foot pressure between groups.

		With Pelvic-Tilt Imbalance (*n* = 25)			Without Pelvic-Tilt Imbalance (*n* = 16)			*p*
Erector-spinae-muscle activity	Flexion peak ratio	0.8	±	0.133	0.72	±	0.18	0.307
	Re-extension peak ratio	0.87	±	0.08	0.83	±	0.13	0.435
Foot pressure	Rt/Lt ratio	0.94	±	0.05	0.93	±	0.04	0.307
	Rt. front/back ratio	0.81	±	0.14	0.86	±	0.12	0.347
	Lt. front/back ratio	0.81	±	0.16	0.79	±	0.11	0.361

Values are mean ± SD, Rt: right, Lt: left.

## Data Availability

We can’t provide data because of privacy.

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
