# Peer review of "Effects of Pelvic-Tilt Imbalance on Disability, Muscle Performance, and Range of Motion in Office Workers with Non-Specific Low-Back Pain"

_healthcare, 2023, doi:10.3390/healthcare11060893_

Round 1

Reviewer 1 Report

This study is good study to investigate the effect of left-right imbalance of the pelvis on back pain and various physical variables. In this study, it was found that back pain patients with left/right pelvic imbalance had more severe disability and a difference in the left/right rotation ratio of the hip joint than back pain patients without pelvic imbalance. It seems to be a good research material to consider when treating patients in clinical practice. Nevertheless, I think it will be a better study if a few supplements are made.

Please unify terminology throughout the study. For pelvic alignment, it is better to unify the left/right imbalance of the pelvis.

1. Introduction

  1) It would be good to describe a little more research on the left/right imbalance of the pelvis and back pain, and to supplement the differences with this study. The current content is too simplistic.

  2) “Several clinical studies have described the relationship between changes in the lumbar curve and those in the trunk and hip muscles according to the anteroposterior tilt of the pelvis.” You need a reference to that sentence.

2. Method

  1) Please describe the period during which the research subjects were recruited.

  2) Please describe the method of recruiting research subjects.

3. Discussion

  1) There is something about foot pressure in the results of the study, but it is omitted from the discussion. A variable that does not differ between the two groups when referring to the results of the study. Please also discuss why there was no difference.

  2) There are many variables in this study. Among each variable, there are variables that have not been discussed. Example: Items in Table 2 were not discussed in their entirety. Please discuss further.

Author Response

Reviewer 1

This study is good study to investigate the effect of left-right imbalance of the pelvis on back pain and various physical variables. In this study, it was found that back pain patients with left/right pelvic imbalance had more severe disability and a difference in the left/right rotation ratio of the hip joint than back pain patients without pelvic imbalance. It seems to be a good research material to consider when treating patients in clinical practice. Nevertheless, I think it will be a better study if a few supplements are made.

Please unify terminology throughout the study. For pelvic alignment, it is better to unify the left/right imbalance of the pelvis.

response: Terminology was unified.

  1. Introduction

1) It would be good to describe a little more research on the left/right imbalance of the pelvis and back pain, and to supplement the differences with this study. The current content is too simplistic.

response:  As a result of a comprehensive analysis of 24 studies in a systematic review by Yoon WG and others, it was found that there was a significant relationship between left and right pelvic imbalance and back pain

2) “Several clinical studies have described the relationship between changes in the lumbar curve and those in the trunk and hip muscles according to the anteroposterior tilt of the pelvis.” You need a reference to that sentence.

response: This sentence is a summary of the results of several studies described above. References are given in the preceding sentences.

  1. Method

1) Please describe the period during which the research subjects were recruited.

response: This study analyzed data collected from May 2018 to December 2018 for the health of workers at the S Company Musculoskeletal Therapy Center located in Seoul.

2) Please describe the method of recruiting research subjects.

response: Subjects were recruited through the internal bulletin board.

  1. Discussion

1) There is something about foot pressure in the results of the study, but it is omitted from the discussion. A variable that does not differ between the two groups when referring to the results of the study. Please also discuss why there was no difference.

response: Contrary to the hypothesis of this study, there was no significant difference in trunk muscle activity and foot pressure between the two groups. The difference in the left and right pelvic tilt can affect the mobility and balance of the trunk and lower extremities. However, this is not determined only by the tilt of the pelvis and is balanced by various compensatory actions. For this reason, it is thought that there was no difference between the two groups in trunk muscle activity and plantar pressure distribution.

  2) There are many variables in this study. Among each variable, there are variables that have not been discussed. Example: Items in Table 2 were not discussed in their entirety. Please discuss further.

response:  There was no significant difference between the groups in the weekly computer use time and the weekly exercise time. It is thought that these results appeared because the weight distribution was not considered when using a computer and the specific types of exercise were not investigated. Future studies will need to improve this aspect.

Reviewer 2 Report

The meaning of the sentence below in the abstract is unclear.-This study is a cross-sectional comparison study/in-house physical therapy clinic.

Please indicate references on the reliability and validity of the Pelvis alignment imbalance measuring device (Formetric 4D device). In addition, a detailed description of the measurement method is required. Please add an explanation of the subject's posture (measured in a standing position?), how to apply the equipment, etc. It is appropriate to express it with a picture.

Why are those who used a computer for less than 74 hours per week excluded? What caused 72 hours to be the standard?

Who did the subject evaluation? Are the evaluators sufficiently skilled professionals?

Author Response

1) The meaning of the sentence below in the abstract is unclear.-This study is a cross-sectional comparison study/in-house physical therapy clinic.

response: This study is a cross-sectional comparison study => This study is a cross-sectional study

in-house physical therapy clinic. => in physical therapy clinic.

2) Please indicate references on the reliability and validity of the Pelvis alignment imbalance measuring device (Formetric 4D device).

response: Degenhardt BF, Starks Z, Bhatia S. Reliability of the DIERS Formetric 4D spine shape parameters in adults without postural deformities. BioMed research international. 2020 Feb 13;2020.

3) In addition, a detailed description of the measurement method is required. Please add an explanation of the subject's posture (measured in a standing position?), how to apply the equipment, etc. It is appropriate to express it with a picture.

response: The subject's evaluation posture was explained in the text. I don't have it because I didn't take a picture. sorry.

4) Why are those who used a computer for less than 74 hours per week excluded? What caused 72 hours to be the standard?

response: This study investigated the risk factors of back pain according to pelvic imbalance in office workers. It was to select subjects who sat for long hours in consideration of the characteristics of office workers.

5) Who did the subject evaluation? Are the evaluators sufficiently skilled professionals?

response: All evaluations were conducted by a physical therapist with 8 years of clinical experience, regardless of the study design.

Reviewer 3 Report

Back pain affects a relatively large group of people. Any research aimed at explaining their causes can improve the health of people affected by this problem.

In the following work, the authors examined a group of office workers with pain. The research was aimed at examining whether there is a relationship between pain and changes in the anatomical structure.

The research was carried out and developed in a manner typical for this type of study. The authors presented the problem, discussed the research methodology, presented the results and conducted a discussion.

Minor remarks concern the conclusions. In the opinion of the reviewer, the authors did not fully respond to the assumptions of the work in their conclusions. This would require a correction to answer the question of what is the relationship between the presence of pelvic tilt imbalance and the various physical risk factors for low back pain discussed in the introduction.

In the opinion of the reviewer, the article can be accepted for publication.

Author Response

Back pain affects a relatively large group of people. Any research aimed at explaining their causes can improve the health of people affected by this problem. In the following work, the authors examined a group of office workers with pain. The research was aimed at examining whether there is a relationship between pain and changes in the anatomical structure. The research was carried out and developed in a manner typical for this type of study. The authors presented the problem, discussed the research methodology, presented the results and conducted a discussion.

Minor remarks concern the conclusions. In the opinion of the reviewer, the authors did not fully respond to the assumptions of the work in their conclusions. This would require a correction to answer the question of what is the relationship between the presence of pelvic tilt imbalance and the various physical risk factors for low back pain discussed in the introduction.

response: In the conclusion, occupational characteristics were added and explained.

In the opinion of the reviewer, the article can be accepted for publication.